# Study on the Mechanical Properties of Fly-Ash-Based Light-Weighted Porous Geopolymer and Its Utilization in Roof-Adaptive End Filling Technology

**DOI:** 10.3390/molecules26154450

**Published:** 2021-07-23

**Authors:** Luchang Xiong, Bowen Fan, Zhijun Wan, Zhaoyang Zhang, Yuan Zhang, Peng Shi

**Affiliations:** 1Key Laboratory of Deep Coal Resource Mining (CUMT), Ministry of Education of China, Xuzhou 221116, China; lchxiong@cumt.edu.cn (L.X.); ts20020012a31tm@cumt.edu.cn (B.F.); zhyzhang@cumt.edu.cn (Z.Z.); ts19020038a31tm@cumt.edu.cn (P.S.); 2School of Mines, China University of Mining & Technology, Xuzhou 221116, China

**Keywords:** FBLPG, porous material, porous structure, mechanical property, filling and plugging

## Abstract

This paper aims to study the porous structure and the mechanical properties of fly-ash-based light-weighted porous geopolymer (FBLPG), exploring the feasibility of using it in roof-adaptive end filling technology based on its in-situ foaming characteristics and plastic yielding performance. A porous structure model of FBLPG during both the slurry and solid period was established to study their influence factor. In addition, this study also built a planar structure model in the shape of a honeycomb with bore walls, proving that the bore walls possess the characteristics of isotropic force. FBLPG shows a peculiar plastic yielding performance in the experiment where its stress stays stable with the gradual increase of the deformation, which can guarantee the stability of a filling body under the cycled load from the roof. At the same time, the in-situ foaming process combined with the unique filling technique can make the FBLPG filling body fully in contact with the irregular roof. This roof-adaptive end filling technology makes it a successful application in plugging the 1305 working face, which avoids problems of the low tight-connection ratio and secondary air-leakage channel resulted from the traditional filling technology, effectively improving coal production in terms of safety and high efficiency.

## 1. Introduction

With the increasing depletion of coal resources in eastern China, the center of the Chinese coal industry has gradually moved to the northwest of China. With the shift of coal industry center comes coal-electricity integration, which contributes to the coal-electricity joint ventures [1,2], bringing a lot of opportunities for the development of Northwestern China and problems regarding energy, environment and economy to this place. Located too far inland and covering a vast area of wilderness, Northwestern China is characterized by drought, ecological fragility and difficulty in resource exploitation [3]. In terms of coal mining and exploitation for electricity generation, it is exceedingly difficult to transfer coal resources exploited there to Eastern China for industrial utilization due to the harsh environment of Northwestern China [4], resulting in pithead plant, which is built in the places near the coal mining places for the convenience of electricity generation with the coal resources. Instead of transferring coal resources from the west to the east, transferring electricity generated in the pithead plant is easier, which is greatly advocated by the government. However, with the spread of this mode of coal-electricity integration comes the big issue of dealing with the fly ash, a type of solid waste from the process of electricity generation in a pithead plant [5,6].

In the coal mining process, longwall mining easily leads to the formation of an unstable triangle structure at the end of the roadway, resulting from a slow collapse of the roof. If there are no measures taken to plug the triangle structure, there will be problems appearing, such as the residual coal in the mined-out area will get oxidized and there will be air loss in the working face [7,8], bringing serious harm to the safety and efficiency of coal mining [9]. There are two kinds of plugging measures usually used in coal mining, including the use of brattice to keep off the wind and end filling. The former way can be achieved at a very low cost, with a poor effect of plugging; while through the latter way, the channels will be effectively plugged, promoting the safe coal mining with high efficiency, even though the filling technology is quite complicated and has a higher cost [10]. Therefore, it is worth promoting the end-filling technology. The usual end-filling materials mainly include OPC cement and some chemical agents like Malisan. The former way, although with low cost, has a poor roof-adaptive performance, and easily leads to new air leakage channels after the filling body is crushed and destroyed by the broken overlying strata. The latter way costs a lot with serious heat release, which makes it impossible to use in mines. Therefore, figuring out a new way of end-filling with reasonable cost, the technical feasibility and optimal effect becomes the issue needing to be tackled in the field of coal mining.

The fly ash is the collected small ash particles from flue gases in the coal combustion process, and the study on fly ash abroad began in the early 20th century [11,12], while the study on its application can be traced back to the 1950s [13,14]. With the increasingly deep study on the fly ash and more and more mature technology for its application, fly ash has become a raw material for construction with the price of 200 to 400 Yuan per ton. However, in Northwestern China, there are problems in the application of fly ash, such as the conflict between large production and low market demand, and the severe pollution brought by it, etc. Actually, the fly ash, experiencing the calcination process in a blast furnace in a power plant, can be effectively utilized in industry for its volcanic ash activity, which draws attention from most scholars and thus is studied by them for its industrial use, including the study on its particle size, activation technology and so on [15,16]. At present, fly-ash-based geopolymer has been widely applied in many areas of daily life, ranging from the construction of the road, airport runways and floors, even in fields like 3D printing, hazardous waste immobilization, and molecular sieve [17,18]. As a green material, the application of a fly-ash-based geopolymer with high-efficiency in the industry is worthy of further study. In-situ utilization of fly ash from the coal-electricity integration power plant was proposed. In this mode, the fly ash, which is the main material for the preparation of mining functional materials, is used in the daily production activities in coal mining, including the filling and plugging of the subsidence areas and the channel ends, grouting in the mined-out area and the broken surrounding areas and so on. This mode of coal-electricity is to use the fly ash in coal mining as an important resource through useful technology with low cost [19,20]. Our research group, together with Coal and Electricity Company, Ltd, belonging to the China Energy Investment Group, conducted research from 2018 to 2020 on the comprehensive utilization of fly ash from a mine pit in the number one coal mine in Dananhu Lake, for the purpose of providing solutions to the problem of utilization of fly ash in a power plant integrating coal and electricity at a large scale. The research work includes site investigation, material preparation, equipment selection, process design, industrial test and effect analysis and so on, through which the fly-ash-based functional materials for mining are developed and prepared, and they are successively applied in the areas, such as surrounding rock support and fire prevention and extinguishing in mined-out areas.

A fly-ash-based geopolymer has been a research focus in recent years, which can partly or fully replace the Portland cement material that has a high emission of CO_2_ [21,22]. Further study on the activity and the reaction mechanism of fly ash can help improve the reaction degree of the materials and contribute to energy saving and emission reducing [23,24,25]. FBLPG is the combination of a fly ash geopolymer and foam materials. There are two kinds of foaming technologies, including precast foaming and chemical foaming. In the former way of foaming, a special foaming equipment is needed to prepare and mix the foams, which means that this way of foaming relies highly on the equipment. However, chemical foaming is widely applied due to its simple preparation [26]. In the chemical foaming, the foaming of materials is achieved through foaming agents such as aluminum powder, calcium bicarbonate and hydrogen peroxide, and the foaming can be transferred from one place to another because of the time delay of these foaming agents [27]. There are a lot of studies on the application of FBLPG in coal mining, yet there are few studies on its application in the plugging of the channel end. It can be a good idea to use FBLPG in the filling and plugging operation of the end of the mining, realizing the full plugging effect through its in-situ foaming properties. There have been a lot of studies in the metal foam fields on the related experiments involved in the foaming process, including the studies on the foaming mechanism of the porous materials, structure stability and mechanical properties [28,29]. However, there are few theoretical studies on the FBLPG that belong to the inorganic nonmetallic field, and there is a lack of application cases to support the theory.

Based on the previous studies, this paper proposes the roof-adaptive end filling technology, focusing on the peculiar structure and mechanical properties of FBLPG. In addition, good experimental effects have been achieved in the test where the FBLPG is applied for the plugging of the end of the channels in mining. This kind of roof-adaptive end filling technology using FBLPG can be a way to promote the in-situ utilization of the fly ash in power plants of coal and electricity integration in Northwestern China, providing the theoretical guidance and application experience for the end plugging of the working face in longwall mining with good application effect and low cost.

## 2. The Modeling of the Basic Structure of FBLPG Materials

### 2.1. The Preparation of the FBLPG Materials

FBLPG is a light-weighted porous material prepared based on the fly ash geopolymer through a combination of different foaming methods. There are now two kinds of foaming techniques commonly used. The first one is to prepare the foam first before it is put into the slurry fly ash geopolymer to form the light-weighted porous materials through high-speed stirring or air compress of foaming agents made by animal and plant protein. The second one is to put hydrogen peroxide, aluminum powder and sodium bicarbonate into the slurry fly ash geopolymer to foam. Despite that there is a big difference in the foaming method between these two foaming techniques, there are no differences in its effect on the porous structure and mechanical properties of the porous materials. This study focuses on the FBLPG material through the second foaming technique, of which the ratio of the preparation is shown in Table 1.

In the process of preparing the materials, fly ash and cement clinker serve as basic cementitious materials in the reaction. Sodium hydroxide and sodium sulfate are used to excite the potential activity of fly ash. Calcium silicate is formed through the chemical reaction between calcium oxide and activated silica to strengthen the activity of fly ash in the initial reaction stage. Calcium stearate serves as the foam stabilizer to reduce the tension on the surface of the babbles.

### 2.2. The Model for the Globular Unit Structure of FBLPG Slurry

In the preparation process, the dry materials will be intimately mixed, and then the water will be added into the mixed dry materials and stirred for 2 min. Lastly, the prepared hydrogen peroxide will be put into the mixed materials and stirred for 30 s, and the slurry FBLPG is prepared. Then the slurry will be put into the mold to wait for the foaming process. When the slurry is in the process of foaming, under basic conditions, the hydrogen peroxide is decomposed into oxygen, which leads to bubbles of large and small sizes randomly distributed in the slurry. The size and the distribution of the bubbles are affected by the stirring result of the slurry. The slurry FBLPG is made up of materials of three physics states, which are respectively solid, liquid and gas. The picture of the ideal structure model of the slurry is shown in Figure 1. To establish this structure model, the sample was incised, and the pore structure data were obtained by binarization processing of the profile, including pore quantities, size and distribution, and this structure data obtaining method can be found in the work of Xiong in 2019 [20]. All the data were employed by Rhino 6.0 to establish this model. The outer layer is the matrix, and the porous structure in the inner wall is full of bubbles. There is calcium stearate powder of very small size between the outer layer and the inner wall, keeping the pore structure stable.

### 2.3. The Model for the Three-Dimensional Structure of Solidified FBLPG

The FBLPG slurry is cured at 20 °C with a humidity of 98%. In the continuous hydration reaction, the water of a dissociative state participated directly in the reaction and became crystalline. The hole in the FBLPG slurry is supported by the air pressure in the bubbles. In this reaction process, the small bubbles merge and gradually become large ones. After this curing process, the FBLPG slurry is cured and there are stable pore structures formed inside the FBLPG due to the substrate consolidation, making FBLPG a material with a lot of pores. In this study, the model for the solidified FBLPG was established by using Rhino software. In this model, according to some parameters like the pore size and number of FBLPG measured, the bubbles were randomly distributed in the substrate. The modeling was completed after the bubbles were separated from the matrix, and the model is shown in Figure 2.

## 3. Mechanical Properties Study of FBLPG

### 3.1. Analysis of the Factors Influencing Distribution Characteristics of FBLPG Pores

In a great number of sample observations, it has been found that the pore size of FBLPG shows a gradually enlarging variation trend from the bottom to the top. Through a theoretical analysis, it can be found that this variation trend is related to the formation and curing process of the bubbles when the FBLPG is in a slurry state. The relationship between the gas volume and pressure can be expressed by the equation as below. When the temperature of the environment and mol of the gas remain constant, the gas volume is inversely proportional to the pressure.
(1)PV=nRT

In the equation, P represents the gas pressure and V stands for the gas volume. n represents the number of the gas molecule, R is a constant and T stands for the temperature of the environment.

The pressure faced by the bubble in the FBLPG slurry can be expressed in the following equation. It is related to the density of the slurry and the depth of the bubbles.
(2)P=ρgh
(3)$$V=\frac{\mathrm{nRT}}{\rho \mathrm{gh}}$$
where ρ stands for the density of the slurry, and g represents the gravitation constant.

As the equation shows, the micro-aggregates of hydrogen peroxide are uniformly distributed in the slurry under the condition that the slurry is uniformly stirred. Due to the homogeneity of the gel, the density of the slurry can be considered as consistent, and the relative number of the mols of the hydrogen peroxide in different positions is also consistent. Therefore, in the forming mold, the nearer the bubbles are to the bottom of the mold, the smaller their sizes; the closer the bubbles are to the surface, the larger their sizes. However, since the size of the whole mold is small, the variation of the sizes of the bubbles can be ignored to some extent.

Through cutting the FBLPG sample, the internal section of the whole sample is obtained, as is shown in Figure 3. It can be seen that the bore walls of the pores show a sphere shape and honeycomb shape with hexagonal structure. The former shape appears when there is 3% hydrogen peroxide, and the latter arises when there is 7% hydrogen peroxide, which shows that the gas volume increases gradually with the increase in the hydrogen peroxide, and that the bubbles become larger and larger with the reaction going on. As a result, the triangle liquid channel is formed due to the extrusion of the bubbles, and when it gets dried, it becomes a solid bore wall of hexagonal structure. To increase the possibility of obtaining large pores and reducing the density of the material, there should usually be 6% to 7% of FBLPG, and the bore wall is usually in the shape of a hexagon honeycomb.

### 3.2. A Force Analysis of the Bore Walls in the Hexagon Honeycomb Cell Structure

The pore structure of the FBLPG can be considered as the unit structure of the hexagon, which is an independent unit cell of the honeycomb structure. This study analyzes the force in the hexagonal structure of the unit cell through parameter calculation of the mechanical properties, for the purpose of studying the bearing capacity and stability of light-weighted porous materials. The model made by Fu and Zhao is used to achieve the purpose [30,31]. The force model for the bore walls of the hexagon honeycomb structure is a simplified Y model, as shown in Figure 4. Considering the stress in the x and y direction, the force matrix and the balanced state, the elastic modulus and the Poisson’s ratio holding a large influence on the FBLPG deformation are calculated. Here, this paper will only present the calculations of the mechanical parameters based on the stress in the x direction and its torque. Since the calculation based on the force in the y direction is similar to that in the x direction, the calculating results will be given without presenting the calculating process.

The elastic modulus of the FBLPG is
(4)$$P_{x}=\sigma_{x}b\left(h+l\sin\theta \right)$$

Assuming that there is no rotation angle at point A,
(5)$$M_{x}=\frac{P_{x}}{2}l\sin\theta$$
the deflection of AB:(6)$$\delta_{\mathrm{AB}1}=\frac{M_{x}l^{2}}{2\mathrm{EI}}-\frac{P_{x}\sin{\theta l}^{3}}{3\mathrm{EI}}=\frac{P_{x}\sin{\theta l}^{3}}{E_{s}\mathrm{bd}}$$
the elongation of AB:(7)$$\delta_{\mathrm{AB}2}=\frac{P_{x}\mathrm{lcos}\theta }{E_{s}\mathrm{bd}}$$
the deflection of BC:(8)$$\delta_{\mathrm{BC}}=\frac{P_{x}l^{3}\sin\theta }{E_{s}{\mathrm{bd}}^{3}}$$
the elongation of BC:(9)$$\delta_{\mathrm{BC}}=\frac{P_{x}\mathrm{lcos}\theta }{E_{s}\mathrm{bd}}$$
equivalent strain of x direction:(10)$$\varepsilon_{x}=\frac{2\left(\delta_{\mathrm{AB}1}\sin\theta +\delta_{\mathrm{AB}2}\cos\theta \right)}{2l\cos\theta }=\frac{P_{x}\sin^{2}{\theta l}^{3}}{E_{s}\mathrm{bl}\cos{\theta d}^{3}}(1+\cot^{2}\theta \frac{d^{2}}{l^{2}})$$
equivalent strain of y direction:(11)$$\varepsilon_{y}=\frac{2\left(\delta_{\mathrm{AB}2}\sin\theta +\delta_{\mathrm{AB}1}\cos\theta \right)}{h+l\sin\theta }=-\frac{P\sin\theta \cos{\theta l}^{3}}{E_{s}b(h+l\sin\theta )d^{3}}(1-\frac{d^{2}}{l^{2}})$$
the Poisson’s ratio and Elastic modulus of x direction: (in which: β—h/l; Es—Elastic modulus)
(12)$$\mu_{\mathrm{mx}}=-\frac{\varepsilon_{y}}{\varepsilon_{x}}=\frac{\cos^{2}\theta }{\left(\beta +\sin\theta \right)\sin^{2}\theta }\frac{1-\frac{d^{2}}{l^{2}}}{1+\cot^{2}\frac{d^{2}}{l^{2}}}$$
(13)$$E_{\mathrm{mx}}=\frac{\delta_{x}}{\varepsilon_{y}}=E_{s}\frac{d^{3}}{l^{3}}\frac{\cos\theta }{\left(\beta +\sin\theta \right)\sin^{2}\theta }\frac{1}{1+\cot^{2}\theta \frac{d^{2}}{l^{2}}}$$

The Poisson’s ratio and elastic modulus of the y direction and elastic modulus can be calculated in the same way:(14)$$\mu_{\mathrm{my}}=\frac{\left(\beta +\sin\theta \right)\sin\theta }{\cos^{2}\theta }\frac{1-\frac{d^{2}}{l^{2}}}{1+\left(2{\beta \sec}^{2}\theta +\tan^{2}\theta \right)\frac{d^{2}}{l^{2}}}$$
(15)$$E_{\mathrm{my}}=E_{s}\frac{d^{3}}{l^{3}}\frac{\left(\beta +\sin\theta \right)}{\cos^{3}\theta }\frac{1}{1+\left(2{\beta \sec}^{2}\theta +\tan^{2}\theta \right)\frac{d^{2}}{l^{2}}}$$

When the bore wall is in the shape of a hexagon, and β = 1, θ = $\frac{\pi }{6}$, the equation can be simplified as follows:(16)$$E_{\mathrm{mx}}=E_{\mathrm{my}}=\frac{4}{\sqrt{3}}E_{s}\left(\frac{1}{1+3\frac{d^{2}}{l^{2}}}\right)\frac{d^{3}}{l^{3}}$$
(17)$$\mu_{\mathrm{mx}}=\mu_{\mathrm{my}}=\frac{1-\frac{d^{2}}{l^{2}}}{1+3\frac{d^{2}}{l^{2}}}$$

Through the calculation, it can be seen that the Poisson’s ratio and elastic modulus of the x direction is highly consistent with those in the y direction, which shows that the bore walls of FBLPG possess the characteristics of isotropic force.

### 3.3. An Analysis of the Plastic Yielding Performance of FBLPG

To study the bearing capacity of FBLPG, the universal testing machine of type WDW-300 is used in the uniaxial compressive strength test with invariable displacement loading speed. The fly-ash-based geopolymer sample that is not foaming is cast in a rectangular mold of 70.7 mm, while the FBLPG sample is cast in a rectangular mold of 100 mm. The details on the testing process are shown in Figure 5. In the test, it can be easily observed that the fly-ash-based geopolymer sample that is not foaming shows regular damage form as in rock and concrete. It seems to have a strong bearing capacity, up to 20 MPa, even though it shows a weak deformation resistance. When it reaches the yielding limit, distinct cracks arise in the sample and then it gets collapsed. However, there are few cracks appearing in the FBLPG sample, and only a small collapse arises at the end of the channels. There was a great deformation in the whole sample.

FBLPG sample shows better plasticity than the regular geopolymer in the bearing process and has the typical plasticity belonging to porous materials. There are stress–displacement curves of the foaming and not-foaming fly-ash-based geopolymer in Figure 6. It can be clearly seen that the materials hold distinct plasticity in some phases and show rapid decline in the stress after experiencing their peaks in the stress. For FBLPG, its stress remains constant with the deformation change after the short plastic phase.

This phenomenon is closely related to the unique hexagon honeycomb structure of FBLPG. The loading procedure of FBLPG consists of three main stages, as is shown in Figure 7. In the first stage, after the cutting, the incomplete pores were damaged first. As the FBLPG surface began to come in contact with the head of the testing machine, the incomplete pores in the surface began to get pressed and gradually collapsed. In this stage, the whole FBLPG sample remained elastic for a short period. In the second stage, the weakness plane was pressed by the force. The weakness plane in the structure and the pore were damaged, resulting in the main fracture. With the loading going on, the micro-fracture began to get closed due to the increasing pressure from the loading. The pores also began to disappear from the surface to the inside. In this stage, the complete pores disappeared because of the pressure from the loading. However, the deformation increased gradually, and there was a very small change in the pressure values. The part of the sample that was on the edge of the roof and the bottom got in contact with the testing machine and got damaged due to the stress concentration. In the plugging of the end, the deformation resistance of FBLPG was the main factor contributing to the plugging effect instead of its bearing capacity. That is to say, it is the plastic yielding performance of FBLPG that plays an important role in the application of FBLPG in the plugging of the end, which will not produce the brittle failure leading to big air leakage channels. Instead, it will only get gradually deformed with the falling of the roof, which can effectively plug the end in the working face.

## 4. Application of FBLPG in the Filling and Plugging in the Roadway End

### 4.1. An Analysis of Air Leakage Channels and Spontaneous Combustion in the Mined-Out Area

Based on the concept of in-situ utilization of fly ash generated in the power plant, the fly ash generated in the power plant of the Dananhu number one coal mine was used to prepare FBLPG, which was applied in the plugging operation in the 1305 working face. According to the determination report on the period of spontaneous coal combustion in this area, CO is the first gas product that appears in the oxidizing process of coal and exists through the whole oxidizing process. Therefore, CO was selected as the first important sign for the spontaneous coal combustion, with its quantity variation and changing rate as indicators for the prediction of the spontaneous combustion trend. As the testing results show, the coal seam mined in this area belongs to a spontaneous combustion seam of type I, with the shortest spontaneous combustion period of 37 days. After the mining of the coal seam, there were a lot of residual coals left in the mined-out areas, and these residual coals exposed to the air would be gradually oxidized, producing a large quantity of CO. As a result, the environment of the working face would be polluted by the CO and the secure operation would be threatened by the spontaneous combustion. There were two surveillance points set on the upper end of the working face, and they were 40 m apart from each other. As the mining in the working face went on, the depth of the surveillance points in the mined-out areas would be increased, and variation trends of the O_2_ concentration and temperature in the surveillance points are shown in Figure 8.

Similarly, there were surveillance points set in the working face and roadway to monitor the variation in the O_2_ concentration and temperature with the depth of the bundle tube. Then the distribution range of the “three zone” of spontaneous combustion of the 1305 working face was determined, as Figure 9 shows, the return air chute is located in the oxidation zone with the width as 27 m. The haulage gate is 50 m, which is the widest in the “three zone” of spontaneous combustion in the mined-out areas.

Although the shortest period of spontaneous combustion of the working face is 37 days, a coal mining operation would be safe at a minimum speed of 1.35 m/d. However, there were problems, such as too much CO left and severe loss of air at the end of the working face, resulting from large air leakage channels formed largely due to the slow collapse of the roof in the two ends of the working face, as is shown in Figure 10. The air in the working face would get into the mined-out areas through the channels, leading to the spontaneous combustion of the residual. Therefore, it is exceedingly necessary to take measures to prevent such potential problems from happening. Usually, the measure for the prevention of such problems is plugging the channels with a windshield cloth or with bricks. Plugging the channels with the former is easy, yet with poor effect, and its plugging effect is deeply influenced by the roof collapse. Plugging the channels by using the bricks put people operating in danger, and there is great possibility for the bricks to collapse, leading to air leakage channels again. In a word, either way of plugging the channels has limitations. Therefore, the FBLPG is used to achieve the purpose of this study.

### 4.2. Roof-Adaptive End Filling Technology

A combination of the overlying strata instability and stress concentration resulting from the mining in the working face at the end of the channels leads to the damage in the walls and irregular roof collapse, which further leads to the poor effect of plugging. It is impossible to plug all the air leakage channels at the end through regular plugging operation with filling bags due to the great difficulty in fully filling the channels. The preparation of FBLPG material needs to experience the foaming process, which contributes to the large increase in its volume and makes it a suitable material for filling and plugging the channels. Therefore, this study proposes to use it for the designation of roof-adaptive end filling technology, which fully takes advantage of FBLPG material to improve the poor effect of filling and plugging achieved by other regular filling materials in the filling and plugging process of the end. The expansion characteristic of FBLPG enables it to get in full contact with the irregular surrounding rocks, improving the compactness of FBLPG at the end and reducing the possibility of the existence of air leakage channels, which can improve the plugging effect.

The procedure of roof-adaptive end filling technology is very easy, with low dependence on devices and is moveable with convenience. Its working principle can be seen in Figure 11. The whole setup consists of three parts: the supporting part, filling bag part and pumping equipment. The supporting part is composed of a triangle supporting platform and a woody baffle plate, providing stability of the filling bag at the end near the mined-out areas. The filling bag serves as a mold for the FBLPG slurry. The pumping equipment consists of an iron mixing cylinder, an HBMD 80 filling pump designed for mining and a high-pressure rubber tube with a maximum pressure bearing of 4 MPa, which is used for the preparation of FBLPG and pumping. It needs to be noted that the upper end of the filling bag is not sealed, ensuring full contact of FBLPG with the roof when it is foaming and expanding, and thus it would become the perfect filling body that fits the space of the channels.

Specifically, the roof-adaptive end filling technology consists mainly of the following five procedures. Firstly, the triangle supporting platform and the woody baffle plate nearing the mined-out areas were installed. Secondly, the filling bag was hung and the bundle tube was buried. Thirdly, the woody baffle plate nearing the channel end and the supporting platform for the roof were installed. Fourthly, the mixing cylinder and filling equipment were installed. After the four preparation procedures, the plugging and filling operation began. The details of the operation can be seen in Figure 12. In the process of filling, the amount of FBLPG needing to be poured was determined by its foaming times. In the experiment, the FBLPG used would foam twice, and therefore, the volume of FBLPG taking up half of the filling space was poured immediately after the adding and full stirring of the hydrogen peroxide. When the FBLPG was stopped to be poured, the filling tube was taken out and the filling mouth was sealed. When FBLPG got into the filling space, it would begin expanding spontaneously and gradually fill the whole space. The effect of the FBLPG plugging and filling was assessed by analyzing the air composition change with the air extracted through the buried bundle tube. Furthermore, the airflow change was also used to evaluate the effect. The details of the evaluation were as follows.

### 4.3. Evaluation of the End Filling and Plugging Effect

The main purpose of plugging the channel end is to prevent the fresh air from going into the mined-out areas. Otherwise, the residual coals in the areas would be oxidized, producing toxic CO, which would accumulate at the end and poses danger to the operation of the mining in the working face. In addition, the large size channels resulting from the collapse of the end would lead to the air loss and thus cause airflow loss. Therefore, CO concentration and the airflow were selected as the two key indicators for the evaluation of the plugging and filling effect. The monitoring point for CO concentration was set at the end of the working face, and the collection of and concentration testing of CO were done through the JSG-7 monitoring system. The airflow speed was measured through the DFA-IV airflow meter by the means of a four-line method. Then, the section areas of the monitoring points in the roadway were measured. Through calculation, the airflow of the monitoring points can be obtained. The arrangement of the monitoring points for the CO concentration and airflow is shown in Figure 13.

To see the effect of the filling and plugging measures on the control over the CO concentration, the CO concentration was monitored before and after the experiment. Before the experiment, the CO concentration was monitored during the days from the 15th to 18th of June in 2019. The filling and plugging operations were done on the 27 June. In the three days from 27th to 30th of June, the CO concentration was monitored again after the experiment. The gas was collected and tested every three hours, of which the details are shown in Figure 14. As the monitoring results show, there was a similar variation appearing in the monitoring of the CO concentration before and after the filling and plugging. That is, the CO concentration was gradually increasing and then kept stable. However, there was a big difference in the peaks of the CO concentration, and the peak of the CO concentration after the filling and plugging was much lower than that before the process. Before the end of filling and plugging, there was a big triangle space for air leakage at the end, resulting from the collapse of the roof in the working face. As a result, the fresh air goes into the mined-out areas, causing the oxidization of the coal and thus the large quantity of CO gas. The CO concentration gradually increased until it reached the peak, which got under control by the filling and plugging measure, which effectively prevented the fresh air from going into the mined-out areas. According to the monitoring results, it can be clearly seen that the filling operation at the end of the working face effectively reduced the CO concentration to less than 24 ppm, which is in the security range in coal mining. That is to say, this filling technique is effective enough to prevent the fresh air going into the mined-out areas at the end and thus reduce the oxidization of coals in the areas, keeping CO concentration in a secure range.

The monitoring results of the airflow in the working face before and after the filling operation is shown in Figure 15. The air goes into the working face through the intake airflow roadway and goes out from the working face through the return airflow roadway. Before the filling operation, there was great air loss along the way, and there was significant reduction in the air loss after the filling measure. The main reason for this phenomenon is that there was a big air leakage channel at the end before the filling operation, which led to great air loss when the air went into the mined-out areas. There was 20% of air loss along the way from the intake airflow roadway to the return airflow roadway. Through the filling and plugging operation at the end of the working face, the air loss was reduced by 5%, and thus, the operation environment was effectively improved in terms of air loss, which is of great significance to ensure safe and efficient coal mining in the working face and reduces the cost of coal mining.

## 5. Conclusions

Through the experiment and on-site application analysis, there are four conclusions drawn, which are as follows.

Firstly, in the foaming process, the pore distribution of FBLPG is related to the depth of the slurry, which directly leads to the uniform variation of pore size in a small range. In terms of the pore size distribution, the pore size of FBLPG gets smaller linearly in the vertical direction, and in the horizontal direction, there is a slight change in the pore size of FBLPG, which distributes evenly in this direction.

Secondly, the pore shape of FBLPG is largely influenced by the amount of foaming agent. When there is less than 7% of the foaming agent, the pores show a spherical shape. When there is more than 7% of the foaming agent, the pores show a shape of the unit structure of a hexagon honeycomb, and the pore walls show a characteristic of isotropic force.

Thirdly, FBLPG shows a plastic yielding performance in bearing the pressure. In the loading procedure, there are three stages: the stage when the incomplete pores in the surface got damaged, the stage when the weakness plane bore the force and the stage when all the pores disappeared due to the loading pressure. The plastic yielding performance of FBLPG makes it an excellent filling body at the end, for it can be slowly deformed by the gradual collapse of the roof without producing big air leakage channels.

Fourthly, the application of FBLPG in roof-adaptive end filling technology can effectively solve the problems of excessive CO concentration at the end and of airflow loss in the working face resulting from the slow and gradual collapse of the roof, ensuring the safe and efficient operation in the coal mine.

The successful application of FBLPG roof-adaptive end filling technology in 1305 working face proves the possibility of industrial promotion. It is worthy to notice that this kind of end-filling technology avoids the problems of the low tight-connection ratio and secondary air-leakage channel resulted from the traditional filling technology, effectively improving coal production in terms of safety and high efficiency.

## Figures and Tables

**Figure 1 molecules-26-04450-f001:**
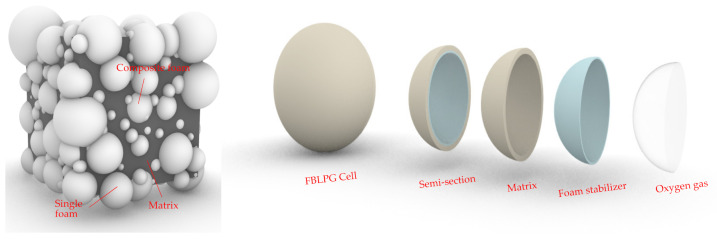
The model for the structure of the FBLPG slurry.

**Figure 2 molecules-26-04450-f002:**
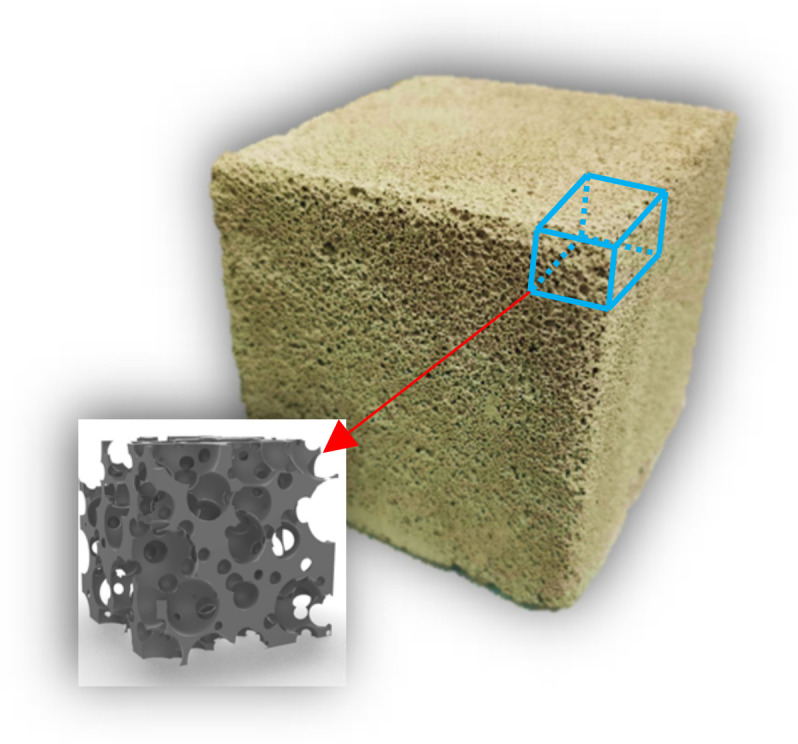
The model for the structure of the solidified FBLPG.

**Figure 3 molecules-26-04450-f003:**
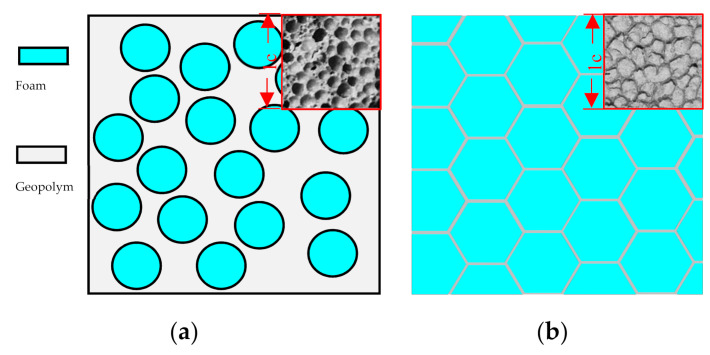
(**a**) The model of a two-dimensional structure with pores of FBLPG at 3% hydrogen peroxide; (**b**) the model of a two-dimensional structure with pores of FBLPG at 7% hydrogen peroxide.

**Figure 4 molecules-26-04450-f004:**
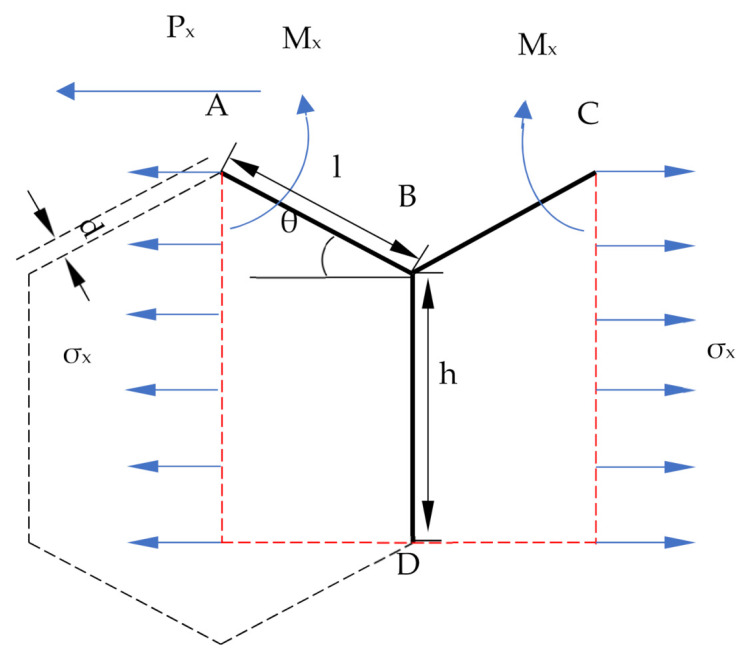
The model for the force in the x direction in the unit structure of FBLPG. D—bore wall depth; l—the length of the bevel edge of the unit cell; h—the height of the vertical side of the unit cell; Mx—matrix in the x direction; σx—stress in the x direction; Px—horizontal force.

**Figure 5 molecules-26-04450-f005:**
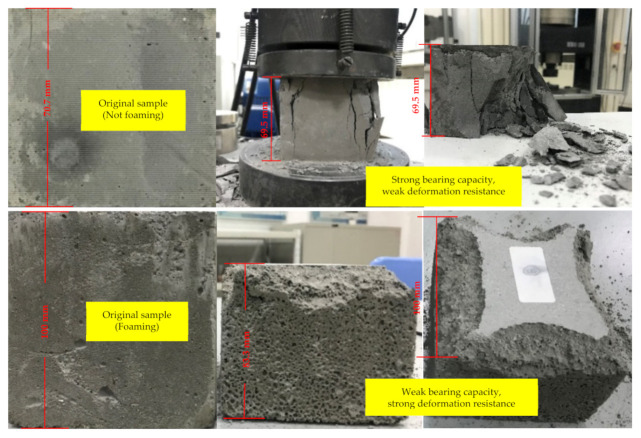
The deformation result of FBLPG and the matrix. Samples.

**Figure 6 molecules-26-04450-f006:**
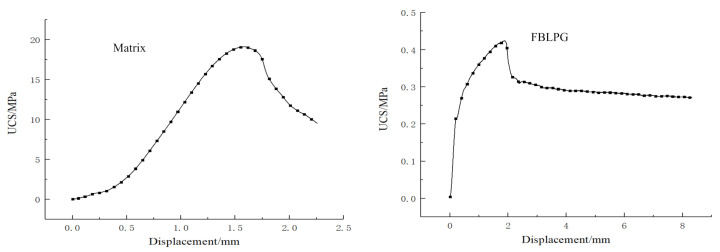
Typical stress–displacement curve of FBLPG (UCS, uniaxial compression strength).

**Figure 7 molecules-26-04450-f007:**
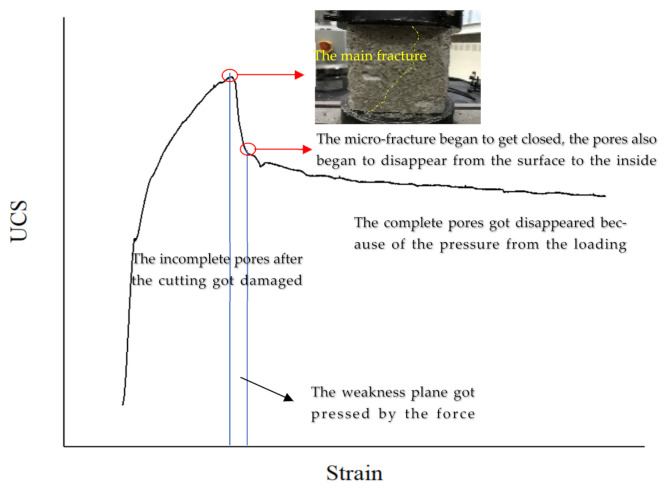
Three stages of the FBLPG damage.

**Figure 8 molecules-26-04450-f008:**
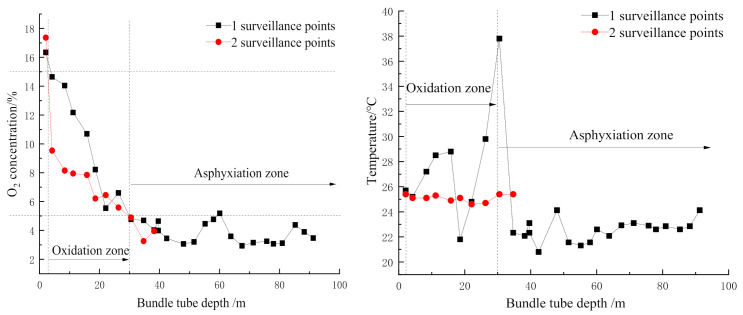
The influence of the bundle tube depth on the O_2_ concentration and temperature in the surveillance points.

**Figure 9 molecules-26-04450-f009:**
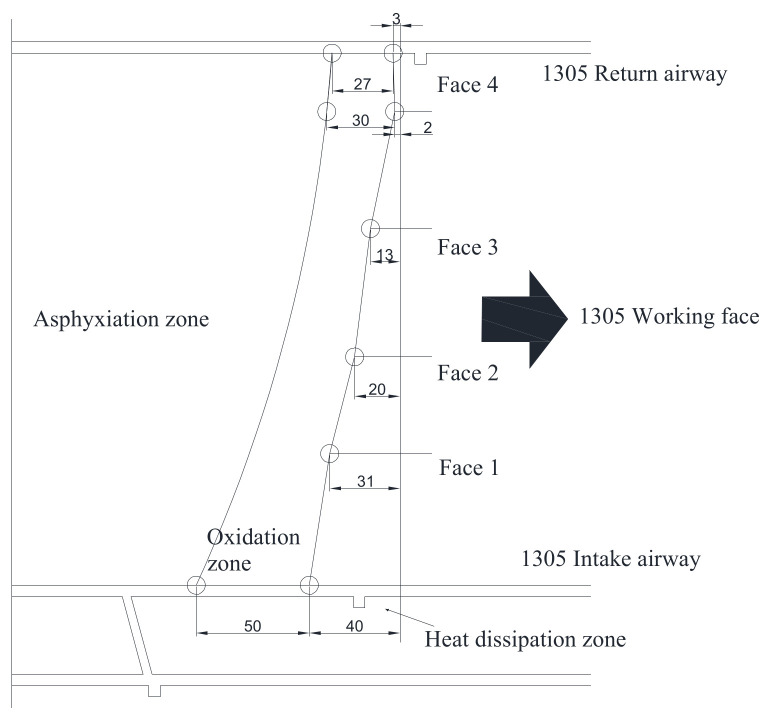
The “three zone” of spontaneous combustion of the 1305 working face.

**Figure 10 molecules-26-04450-f010:**
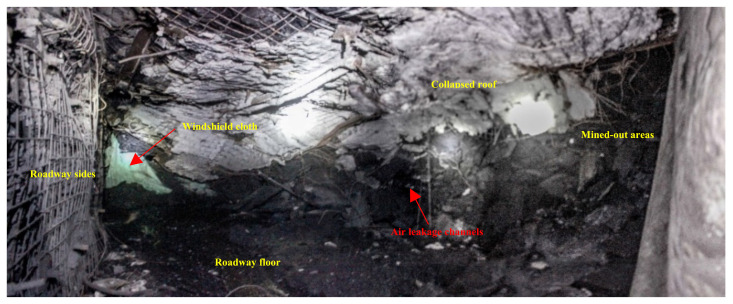
Collapse zone of the lower end of the working face.

**Figure 11 molecules-26-04450-f011:**
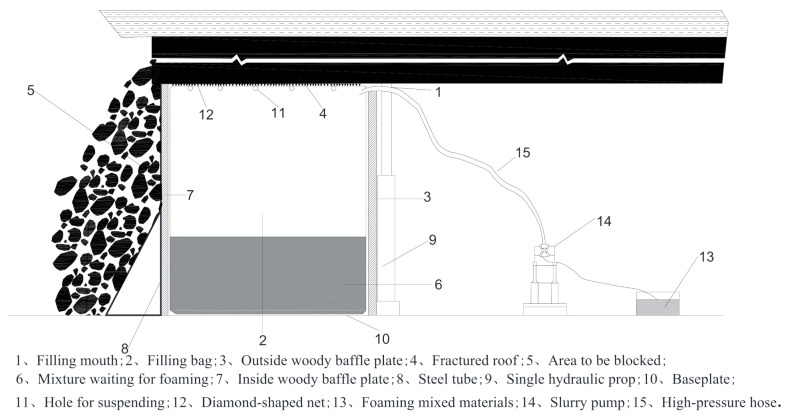
Schematic diagram of roof-adaptive end filling technology.

**Figure 12 molecules-26-04450-f012:**
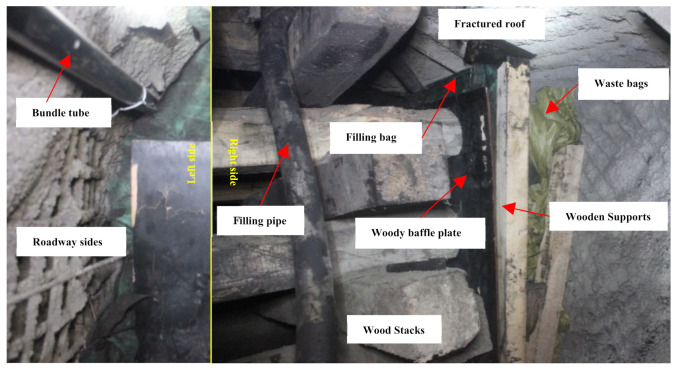
On-site picture of the roof-adaptive end filling technology.

**Figure 13 molecules-26-04450-f013:**
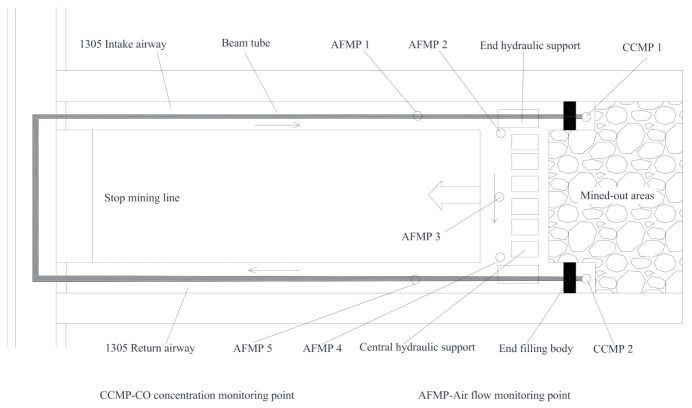
The arrangement of the monitoring points for the CO concentration and airflow in the working face.

**Figure 14 molecules-26-04450-f014:**
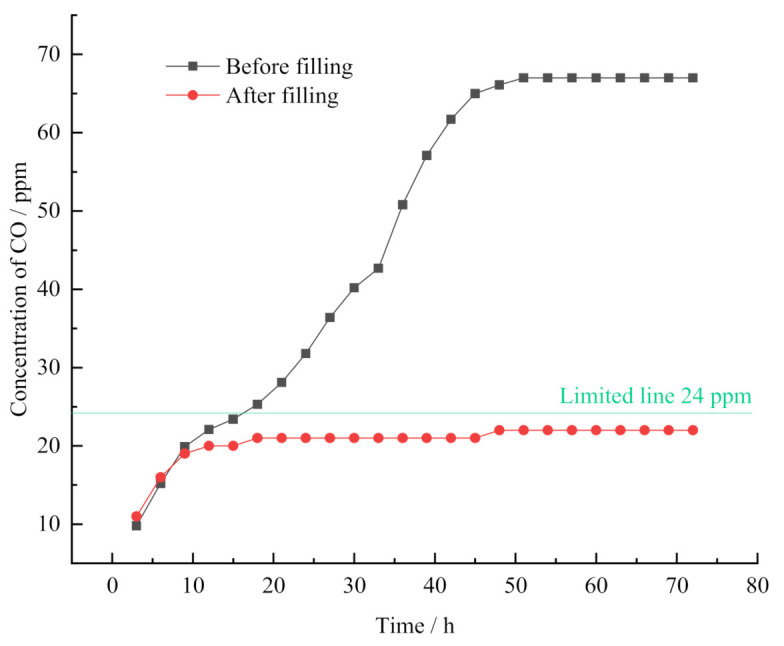
The variation trend of CO concentration in the roadway end before and after the filling operation.

**Figure 15 molecules-26-04450-f015:**
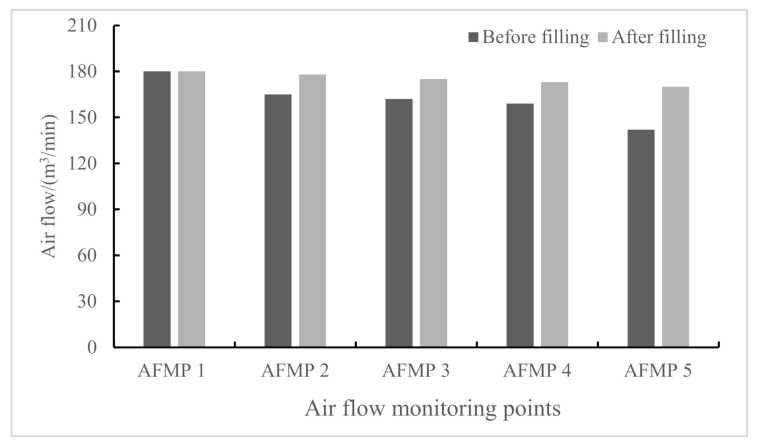
The variation in the airflow in the monitoring points of the working face before and after filling.

**Table 1 molecules-26-04450-t001:** The ratio design for FBLPG.

Raw Material	FA	CC	SH	CM	SS	CS	HP	W
Ratio	0.5	0.5	0.02	0.02	0.01	0.01	0.07	0.36

FA—fly ash, CC—cement clinker, SH—sodium hydroxide, CM—carbon monoxide, SS—sodium sulphate, CS—calcium stearate, HP—hydrogen peroxide, W—water.

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
