# Peer review of "Study on the Mechanical Properties of Fly-Ash-Based Light-Weighted Porous Geopolymer and Its Utilization in Roof-Adaptive End Filling Technology"

_molecules, 2021, doi:10.3390/molecules26154450_

Round 1
Reviewer 1 Report
The MS is not appropriate for publication in Molecules. The MS does not discuss any aspects of materials chemical composition, their physico-chemical properties etc.
Author Response
First of all, thank you very much for the precious time you spent on review our manuscript. Exactly you are right, this work is not focus on the materials chemical composition or physico-chemical properties, but we do think there is some room for this manuscript to be appropriate for this Special Issue “Barrier and Functional Materials from Waste Materials for Pollutants” in Molecules This work is focus on the pore structure and strength performance of the fly ash based light-weighted porous geopolymer, and these characteristics are all very important for the reuse of fly ash, and this paper also contains the industrial application of reused fly ash based geoplymer. So this paper might just meet the aim and requirement that for this special issue “appropriate analytical and technical solutions for the wastes utilization and pollutants control”.
Really appreciate for your time and effort on this reviewing on our work. Hope everything goes well with your work and daily life.
Best regards,
Zhijun Wan

Reviewer 2 Report
The manuscript entitled "Study on the mechanical properties of Fly ash-based light-weighted porous geopolymer and its utilization in roof-adaptive end filling technology" investigated the application of using fly ash-based light-weighted porous geopolymer in roof adaptive end filling by considering their foaming and relatedly mechanical properties. The study and organization of the paper are quite good. I think the paper can be acceptable in its present form but the improvement of discussion quality with previously published reports and the re-written abstract will definitely increase the readability of the paper in the end. And in the second-order, I think some statements should also be used for expressing the novelty and need for such a study in both literature and application in industry.
Author Response
First of all, thank you so much for the comments you made and your affirmation for the manuscript, they are all of great importance to improve the quality of this paper. We have considered for all of your comments carefully and made some related and response for them, meanwhile the related correction have been made in the manuscript with red mark. Hope these correction can meet your requirements. Really appreciate for your precious suggestions for the paper.
The abstract have been rewritten under your suggestion, and related correction has made on the emphasis of the importance and need of this work, both in the rewritten abstract and the conclusion. Hope these change and correction can meet your suggestion.

Reviewer 3 Report
This manuscript investigates the mechanical properties of a porous structure named FBLPG, and studies their potential as in roof-adaptive end filling technology. The topic is interesting, considering in-situ application of fly ash to produce affordable light-weight porous geopolymer structures. Some suggestions for further improvements include:
(1) The English needs to be significantly improved.
(2) Section 2.2, the authors should explain in detail how the model in figure 1 is established and show the solid evidences for the internal structures.
(3) Section 3, the scales of the micrographs need to be shown. The experimental micrographs of the pores are far from uniform, how are these affecting the current analytical model and the lateral results?
(4) What is “UCS”? It needs to be explained in full.
Author Response
First of all, thank you so much for the comments you made for the manuscript, they are all of great importance to improve the quality of this paper. We have considered for all of your comments carefully and made some related and response for them, meanwhile the related correction have been made in the manuscript with red mark. Hope these correction can meet your requirements. Really appreciate for your precious suggestions for the paper.
Response:
As for Opinion 1: the English of this manuscript has been polished by professional after we got the comments.
As for Opinion 2: To establish this structure model, the sample was incised, and the pore structure data was obtained by binarization processing of the profile including pore quantities, size and distribution, and this structure data obtaining method can be found in the work of Xiong in 2019 (See references “Fly Ash Particle Size Effect on Pore Structure and Strength of Fly Ash Foamed Geopolymer” for details). All the data was employed by the Rhino 6.0 to establish this model.
To some extent, cross section of the incised central cubes could reflect the real inner pore structure of FAFG, and therefore, the Nikon D5500 was used to take a high resolution image of the cube side. Pore structure data was analyzed after binarization processing using the open source software “Image Pro Plus”. Raw picture and picture after binarization processing are respectively shown in the following figure where Pore structure and its distribution can be clearly seen.
As for Opinion 3: the scale has been mark in the experimental figures. As for the question for “the experimental micrographs of the pores are far from uniform”, the pore structure of the experimental one does is not the standard hexagon, but the main difference exists in that some of them is pentagon or they are not uniform. This kind of phenomenon might be cause by the fineness of the raw material and curing condition. This structure mode is simplified and ignore these little manmade error, so that we can focus on the main problem. Making a more accurate model to make a near perfect experimental structure is also our target in the further work, and we will do our best for that.
As for Opinion 4: the UCS is short for uniaxial compression strength, and we have made related explain in the manuscript.
Best regards,
Zhijun Wan
